# Effect of a Diet Supplemented with Nettle (*Urtica dioica* L.) or Fenugreek (*Trigonella Foenum-Graecum* L.) on the Content of Selected Heavy Metals in Liver and Rabbit Meat

**DOI:** 10.3390/ani12070827

**Published:** 2022-03-24

**Authors:** Sylwia Ewa Pałka, Ewa Drąg-Kozak, Łukasz Migdał, Michał Kmiecik

**Affiliations:** 1Department of Genetics, Animal Breeding and Ethology, University of Agriculture in Krakow, Al. Mickiewicza 24/28, 30-059 Krakow, Poland; lukasz.migdal@urk.edu.pl (Ł.M.); michal.kmiecik@urk.edu.pl (M.K.); 2Department of Animal Nutrition and Biotechnology, and Fisheries, University of Agriculture in Krakow, St. Spiczakowa 6, 30-199 Krakow, Poland; ewa.drag-kozak@urk.edu.pl

**Keywords:** rabbit, nettle, fenugreek, heavy metals

## Abstract

**Simple Summary:**

Herbs can be a good supplement in an animal diet. With the current increase in the use of herbs and herbal preparations as an animal feed additive, it is very important to monitor the contaminants present in plants, i.e., heavy metals, and to study their content in animal tissues. The toxicity of heavy metals, whether essential or not, depends on several factors including the dose in feed (food), a route of exposure, and sex. Hence, it seems advisable to determine the effect of nettle *(Urtica dioica* L.) leaves and fenugreek *(Trigonella foenum-graecum* L.) seeds in the feed on the content of selected heavy metals in the liver and meat of the rabbit, and determine differences in sex in metal accumulation. The experiment was conducted at University of Agriculture in Krakow (Poland) in the Experimental Station of the Department of Genetics, Animal Breeding, and Ethology. The research material consisted of Termond White rabbits. Until weaning (on the 35th day of life), young rabbits with does were housed in wooden cages. From weaning until the 84th day of life, rabbits were kept in wire metal cages. Three experimental groups were created: the control group (*n* = 20; 10♂ and 10♀) was fed ad libitum with a complete feed. The animals from group N (*n* = 20; 10♂ and 10♀) were fed a complete mixture with added 1% nettle leaves. The rabbits from the group F (*n* = 20; 10♂ and 10♀) were fed with a complete mixture with added 1% fenugreek seeds. The experiment lasted 7 weeks (from 35th to 84th day of the rabbits’ life). All rabbits were slaughtered on the 84th day of age, with an average body weight of 2546 ± 47 g. Samples of liver were taken during the slaughter. Then, the carcasses were cooled for 24 h at 4 °C, and after that time, a sample was taken from the right loin (m. longissimus lumborum) of each carcass. The concentration of heavy metals (Zn, Cu, Ni, Mn, Fe, Pb, Cd) was determined by the atomic absorption spectrometry (AAS). The additives to the feed significantly affected the content of elements in both the liver and the meat of the rabbits (*p* < 0.05). The highest level of the heavy metals, regardless of the used diet, was recorded in the liver. The meat and the liver of rabbits fed with herbal fodder contained less tested metals than in animals fed with fodder without additives. Moreover, more essential metals were found in the liver of rabbits fed with fenugreek than rabbits fed with nettle (*p* < 0.05). In the meat and liver of rabbits, the permissible content of cadmium and lead was not exceeded. Additionally, male livers had a significantly higher content of copper and manganese compared to female livers (*p* < 0.05). This experiment helps to explain the interaction between the heavy metal content of nettle and fenugreek and their content in rabbit meat and liver. The meat (m. longissimus lumborum) and liver of rabbits fed with herbal feed contained fewer tested metals than in animals fed with the feed without additives. Concentrations of toxic metals, i.e., Pb and Cd in liver and meat, were so low that meat consumption does not pose a threat to human health. However, more research is needed to determine how the mechanisms and pathways of heavy metal toxicity act on tissue in which these metals are accumulated.

**Abstract:**

The literature on herbal additives for rabbit feed offers little information on the use of nettle and fenugreek. Both of these herbs are valuable sources of vitamins and minerals. These herbs affect the growth, health, and meat quality of rabbits. They regulate the digestive system, stimulate the appetite, have a positive effect on the functioning of the immune system, and exhibit antibacterial activity. The purpose of the present study was to determine the effect of nettle (*Urtica dioica* L.) leaves or fenugreek (*Trigonella foenum-graecum* L.) seeds in the feed on the content of selected heavy metals in the liver and meat of the rabbit. The rabbits were divided into three groups: group C (*n* = 20; 10♂ and 10♀) was fed ad libitum with a complete feed, N group (*n* = 20; 10♂ and 10♀) was fed a complete mixture with 1% added nettle, and group F (*n* = 20; 10♂ and 10♀) was fed with a complete mixture with 1% added fenugreek. The experiment lasted 7 weeks (from the 35th to the 84th day of the rabbits’ lives). All the rabbits were slaughtered on the 84th day of age, with a body weight of about 2.6 kg. The concentration of heavy metals (Zn, Cu, Ni, Mn, Fe, Pb, Cd) was determined by the atomic absorption spectrometry (AAS). The additives to the feed significantly affected the content of elements in both the liver and the meat of rabbits (*p* < 0.05). The highest level of the heavy metals, regardless of the used diet, was recorded in the liver (*p* < 0.05). The meat (m. longissimus lumborum) and the liver of rabbits fed with herbal fodder contained less tested metals than in animals fed with fodder without additives (*p* < 0.05). Moreover, more essential metals were found in the liver of rabbits fed with fenugreek than rabbits fed with nettle. In the meat and liver of rabbits, the permissible content of cadmium and lead was not exceeded. Additionally, male livers had a significantly higher content of copper and manganese compared to female livers (*p* < 0.05). It is important to study the content of heavy metals in the used animal herbal feed additives and their interaction with each other, as they affect the distribution of elements in tissues and organs.

## 1. Introduction

Rabbit meat is characterized by outstanding taste and nutritional and dietary properties. Next to poultry and fish, rabbit meat belongs to the so-called white meats. Like other white meats, rabbit meat contains little iron (1.1–1.3 mg/100 g of meat) and zinc (0.55 mg/100 g of meat) [1]. It also has low sodium content (37–47 mg/100 g of meat). Excess sodium being one of the main causes of hypertension, this makes it an ideal meat for people at risk thereof. Rabbit meat also contains more potassium (428–431 mg/100 g edible fraction) and phosphorus (222–234 mg/100 g edible fraction) than pork, beef, veal, or poultry [2]. Due to the benefits of eating rabbit meat, efforts continue to improve its quality by using more and more new nutritional components in their diet. Attempts are being made to increase the level of nutrients that have a beneficial effect on the human body and to eliminate those causing negative effects [3,4,5].

There have been many publications on the use of herbs and spices (i.e., nettle and fenugreek) in rabbit nutrition and their effect on growth, health, and meat quality [6,7,8]. Nettle leaves are a source of easily digestible minerals, such as calcium (853–1050 mg/100 g dry weight), phosphorus (50–265 mg/100 g dry weight), iron (2–200 mg/100 g dry weight), sulfur (400 mg/100 g DM), potassium (532–613 mg/100 g DM), and sodium (16–58 mg/100 g DM). In addition, they contain vitamin C (20–60 mg/100 g dry weight), vitamin K (0.16–0.64 mg/100 g dry weight), and B vitamins [9,10]. The high content of chlorophyll (0.6–1% DM) and xanthophylls (0.12% DM) is proof of the presence of large amounts of beta carotene, from which vitamin A is produced. In addition, nettle is rich in organic acids (i.e., formic, acetic, pantothenic) and inorganic (silicic) acid, as well as protoporphyrins, tannins, phytosterols, and glycokinins [10,11,12]. This herb can improve biochemical, hematological, and immunological parameters [13]. Further, nettle boasts anti-allergic properties and has a strong detoxifying effect on the body thanks to its mild diuretic and blood purifying properties [10,11,12].

Fenugreek seeds contain 45–60% carbohydrates, 20–30% proteins rich in lysine and tryptophan, 5–10% oils (lipids), pyridine alkaloids, mainly trigonelline (0.2–0.38%), choline (0.5%), flavonoids, free amino acids (4-hydroxyisoleucine (0.09%), arginine, histidine, lysine), calcium and iron, saponins (0.6–1.7%), glycosides, cholesterol and sitosterol, vitamins (A, B1, C), nicotinic acid, and 0.015% volatile oils (n-alkanes and sesquiterpenes) [14,15,16]. Fenugreek stimulates the body to produce mucus (galactomannans), which allows the removal of allergens from the respiratory system and toxins from the urinary tract, and it is useful both in the prevention and treatment of neurodegenerative diseases and metabolic diseases. Numerous studies have shown that fenugreek can also have anti-inflammatory, analgesic, and antipyretic properties [15,17].

In addition to very important macroelements—calcium (Ca), potassium (K), sodium (Na), magnesium (Mg)—and microelements—iron (Fe), nickel (Ni), zinc (Zn), manganese (Mn), copper (Cu)—herbs also contain toxic metals such as cadmium (Cd) and lead (Pb) [9,18,19]. With the current increase in the use of herbs and herbal preparations as an animal feed additive, it is very important to monitor the contaminants present in plants, i.e., heavy metals, and to study their content in animal tissues. Moreover, it must be highlighted that herb addition may also cause problems in the digestive system and intestine. The toxicity of heavy metals, whether essential or not, depends on several factors including the dose in feed (food), a route of exposure, and sex [20,21]. Hence, it seems advisable to determine the effect of nettle leaves and fenugreek seeds in the feed on the content of selected heavy metals in the liver and meat of the rabbit, and determine differences in sex in metal accumulation.

## 2. Materials and Methods

### 2.1. The Animals

The experiment was conducted at the University of Agriculture in Krakow (Poland) in the Experimental Station of the Department of Genetics, Animal Breeding and Ethology. The research material consisted of Termond White rabbits. Until weaning (on the 35th day of life), young rabbits with does were housed in wooden cages. From weaning until the 84th day of life, rabbits were kept in wire metal cages (2 rabbits per cage). The experiment was conducted in a hall equipped with a lighting installation (14 L:10 D), a water trough installation, and a forced ventilation system. In the experiment, we used 10 does and from each litter, one female and one male were randomly assigned to each group. Three experimental groups were created. The control group (*n* = 20; 10♂ and 10♀) was fed ad libitum with a complete feed. The mixture for this group is presented in Table 1. The animals from the group N (*n* = 20; 10♂and 10♀) were fed a complete mixture with 1% added nettle leaves. The rabbits from the group F (*n* = 20; 10♂and 10♀) were fed with a complete mixture with 1% added fenugreek seeds. The experiment lasted 7 weeks (from the 35th to the 84th day of the rabbits’ lives). The nettle leaves (crude protein: 28%, crude fat: 3.3%, crude fiber: 26%) and fenugreek seeds (crude protein: 28.5%, crude fat: 6.5%, crude fiber: 10.2%) were bought from a feed manufacturer, FHP Barbara Ltd. (Poland). These additives were mashed, mixed with all ingredients, and then pelleted. The metal content of these herbs is presented in Table 2.

### 2.2. The Slaughter

All rabbits were slaughtered on the 84th day of age, with an average body weight of 2546 ± 47 g, after 24 h fasting with the access to water. The processes of slaughter were conducted using the methods described by Blasco et al. [22]. Liver samples were taken during the slaughter. Then, the carcasses were cooled for 24 h at 4 °C, and after that time, a sample from the right loin (m. longissimus lumborum) of each carcass was taken. The experiment was conducted under a permit from the Local Ethics Commission (agreement no 267/2018).

### 2.3. Determining the Concentration of Metals

Samples of meat and liver of rabbits as well as nettle and fenugreek weighing about 5 g were placed in test tubes. They were then pre-mineralized with 10 mL of a mixture of perchloric acid (70% HClO_4_) and nitric acid (65% HNO_3_) in a ratio of 1:3 for about 24 h. The samples were then subjected to thermal mineralization with the Velp 20/26 mineralizer, gradually increasing the temperature from 100 °C to 180 °C for 6–7 h. The resulting clear liquid was then diluted to 10 mL with deionized water. The concentration of metals (Zn, Cu, Ni, Mn, Fe, Pb, Cd) was determined by the atomic absorption spectrometry (AAS apparatus Unicam 929 spectrometer) [23]. The results were read on the standard curve using the standards based on the atomic absorption standards developed at the Weights and Measures Office in Warsaw. Results are shown in milligrams per kilogram of wet weight (w.w.) for muscles and liver, and dry weight (d.w.) for herbs.

### 2.4. Statistical Analysis

The statistical analysis was performed using the SAS package [24]. Statistically significant differences between the means were tested using Tukey–Kramer test at the significance level of *p* < 0.05.

The following linear model was used:*Y_ijk_* = *µ* + *DI_i_* + *SEX_j_* + (*DI* × *SEX*)*_ij_* + *ε_ijk_*
where:*Y_ijk_*—analyzed traits,*μ*—overall mean,*DI_i_*—effect of *i*-th diet (*i* = 1, 2, 3),*SEX_j_*—effect of *j*-th sex (*j* = 1, 2),*(DI × SEX)_ij_*—effect of interaction between diet and gender,*ε_ijk_*—residual effect.

Pearson’s phenotypic correlation coefficients were determined using PROC CORR.

## 3. Results

A statistically significant decrease in the tested heavy metals content was found in the liver of rabbits fed with nettle leaves and fenugreek seeds in comparison with the control group (*p* < 0.05) (Table 3). Fe, Zn, Cu, and Mn showed the highest levels of accumulation in the rabbits’ liver fed with fenugreek (70.34 mg/kg, 32.55 mg/kg, 4.27 mg/kg, and 1.96 mg/kg, respectively) compared to rabbits fed with nettle (49.70 mg/kg, 24.19 mg/kg, 3.40 mg/kg, and 1.39 kg/kg, respectively) (*p* < 0.05) (Table 3). The order of essential metals in meat according to their concentrations was: Zn > Fe > Cu > Ni > Mn (Table 4). In addition, no statistically significant differences were discovered between the content of the examined metals in the meat of rabbits that were fed with nettle leaves and those fed with fenugreek seeds, except for Zn (*p* > 0.05) (Table 4).

The rabbits’ liver contained higher properties of all the examined elements in comparison with their meat (*p* < 0.05) (Table 3 and Table 4).

The analysis of lead content in the meat and liver of the studied rabbit population in all groups showed properties below the sensitivity of the method (Table 3 and Table 4). 

In the case of cadmium, in the present study, its content in the meat was also below the sensitivity of the method, while in the liver, the content of this element ranged from 0.02 to 0.17 mg/kg (Table 3 and Table 4). The highest amount of cadmium was recorded in the liver of rabbits fed with fodder containing nettle (0.17 mg/kg). The analysis showed significant differences in the cadmium content in the liver among the groups: the one fed with nettle (0.17 mg/kg), the one fed with fenugreek (0.02 mg/kg), and the control group (0.03 mg/kg) (*p* < 0.05) (Table 3).

The phenotypic correlation coefficients among metals in the liver and meat are presented in Table 5. Zinc in the liver was positively correlated with copper (rp = 0.36), iron (rp = 0.76), and manganese (rp = 0.61) in the liver and zinc (rp = 0.54), copper (rp = 0.66), iron (rp = 0.36), and manganese (rp = 0.57) in the meat (*p* < 0.05). This metal was negatively correlated with nickel and cadmium in the liver and nickel in the meat (respectively: −0.73, −0.44, and −0.44) (*p* < 0.05). Moreover, a positive correlation was found among copper and manganese in the liver (rp = 0.46) (*p* < 0.05). Correlations among copper in the liver and cadmium in the liver and manganese in the meat were negative (respectively: −0.32 and −0.28) (*p* < 0.05). Nickel in the liver was negatively correlated with iron and manganese in the liver (respectively: −0.65 and −0.65) and zinc, copper, iron, and manganese in the meat (respectively: −0.46, −0.52, −0.39, and −0.49) (*p* < 0.05). This metal was positively correlated with cadmium in the liver (rp = 0.39) and nickel in the meat (rp = 0.40) (*p* < 0.05). Positive correlations were noticed among iron and manganese in the liver (rp = 0.46) and zinc, copper, iron, and manganese in the meat (respectively: 0.39, 0.63, 0.35, and 0.54) (*p* < 0.05). Iron in the liver was negatively correlated with cadmium in the liver and nickel in the meat (respectively: −0.34 and −0.39) (*p* < 0.05). Manganese in the liver was significantly and negatively correlated with cadmium in the liver and nickel in the meat (respectively: −0.30 and −0.40) and positively correlated with zinc, copper, and manganese in the meat (respectively: 0.51, 0.39, and 0.26) (*p* < 0.05). Cadmium in the liver was significantly and negatively correlated with zinc in the meat (rp = −0.31) (*p* < 0.05). Moreover, a positive correlation was found among zinc and copper in the meat (rp = 0.38) (*p* < 0.05). Copper in the meat was significantly and negatively correlated with nickel in the meat (rp = −0.36) and positively correlated with iron and manganese in the meat (respectively: 0.48 and 0.79) (*p* < 0.05). Manganese in the meat was significantly and negatively correlated with nickel in the meat and positively correlated with iron in the meat (respectively: −0.41 and 0.57) (*p* < 0.05) (Table 5).

The phenotypic correlation coefficients among metals in the nettle and liver and rabbit meat are presented in Table 6. Zinc in the nettle was positively correlated with the cadmium in the liver (rp = 0.68) and negatively correlated with iron in the meat (rp = −0.30) (*p* < 0.05). Positive correlations were found among copper in the nettle and manganese (rp = 0.37) and cadmium (rp = 0.44) in the liver (*p* < 0.05). A correlation among copper in nettle and manganese in the meat was negative (rp = −0.40) (*p* < 0.05). Nickel in the nettle was positively correlated with manganese and cadmium in the liver and copper in the meat (respectively: 0.27, 0.52, and 0.27) (*p* < 0.05). This metal was negatively correlated with manganese in the meat (rp = −0.28) (*p* < 0.05). Positive correlations were found among iron in the nettle and cadmium in the liver (rp = 0.63) and manganese in the meat (rp = 0.38) (*p* < 0.05). Manganese in the nettle was significantly and positively correlated with iron in the liver and zinc in the meat (respectively: 0.38 and 0.29) (*p* < 0.05). Cadmium in the nettle was significantly and negatively correlated with manganese and cadmium in the liver (respectively: −0.40 and −0.27) (*p* < 0.05) (Table 6).

The phenotypic correlation coefficients among metals in the fenugreek and liver and rabbit meat are presented in Table 7. Zinc in the fenugreek was positively correlated with the iron in the liver (rp = 0.28) (*p* < 0.05). Positive correlations were found among copper in the fenugreek and zinc in the liver (rp = 0.31) and in the meat (rp = 0.28) (*p* < 0.05). A correlation among copper in the fenugreek and cadmium in the liver was negative (rp = −0.32) (*p* < 0.05). Nickel in the fenugreek was positively correlated with nickel in the liver (rp = 0.29) (*p* < 0.05). Manganese in the fenugreek was significantly and positively correlated with zinc in the liver and in the meat (respectively: 0.47 and 0.28) (*p* < 0.05). Cadmium in the fenugreek was significantly and positively correlated with nickel in the liver (rp = 0.28) (*p* < 0.05) (Table 7). 

In this study, the content of elements with respect to the rabbit’s sex was also analyzed. Male livers had a significantly higher content of copper and manganese (3.95 mg/kg and 1.88 mg/kg, respectively) compared to female livers (3.60 mg/kg and 1.63 mg/kg, respectively) (*p* < 0.05). There were no statistically significant differences in the content of other tested elements (*p* > 0.05) (Table 8).

## 4. Discussion

In the literature on herbal additives in rabbit feed, there is little information about enriching the feed with herbal additives, such as on the content of selected elements in rabbit tissues. 

The study indicated that nettle leaves and fenugreek seeds were a good source of essential metals that are useful for the animal body. The metal contents in the fenugreek seeds and nettle leaves were found in the following decreasing order: Fe > Zn > Mn > Cu > Ni. In this study, nettle was the most abundant source of metals. The trace element with the highest concentration in nettle leaves was iron (380.80 mg/kg). Other authors have also confirmed the high concentration of this element in nettle [9,25]. Unfortunately, herbs can also contain toxic substances such as heavy metals such as lead or cadmium. However, Cd was found in nettle leaves and fenugreek seeds at a concentration of 2.29 mg/kg d.w and 0.48 mg/kg, respectively. The presented findings are similar to the results reported by Hagos and Chandravanshi [18] and Raimova et al. [26]. Based on the statistical analysis, the authors noticed differences in the content of heavy metals between the liver and muscles depending on the feed additive. The essential metal contents in the liver of rabbits fed with nettle and fenugreek was found in the following descending order: Fe > Zn > Cu > Mn > Ni, whereas the order of essential metals in meat according to their concentrations was Zn > Fe > Cu > Ni > Mn. In addition, no differences were found between the content of the examined metals in the meat of rabbits fed with nettle leaves and fenugreek seeds, except for Zn.

The recommended daily consumption of metals such as zinc, copper, iron, manganese, and nickel in an adult human is 12.7 mg/day, 1.6 mg/day, 6 mg/day, 3 mg/day, and 150 μg/day, respectively [27,28,29,30,31]. The consumption of rabbits in the world depends on the culinary traditions of countries. Overall, the consumption of rabbit meat in the EU is 0.5 kg per person a year. Taking into account the average content of these micronutrients in the meat of the tested rabbits, the average annual consumption in the EU and the absorption from food on the level of about a few percent may suggest that their content does not pose a risk to the health of the consumer.

According to our research, the liver contained higher values of all the examined elements in comparison with the meat. These results are consistent with the studies of Cygan-Szczegielniak et al. [32] and Kalafova et al. [33]. A liver is the primary target organ for heavy metals principally because it serves as a store for metals, redistribution, and detoxification. Therefore, a liver is regarded as a more sensitive indicator of environmental pollution than other tissues [34].

In studies on the contamination of food of animal origin, it was found that in the group of toxic metals (mercury, arsenic, lead, and cadmium), cadmium and lead are more harmful, taking into account both a number of exceedances in the permissible content and the scale of the associated risks [35,36,37]. A liver and, to a lesser extent, muscles, apart from micronutrients, also accumulate heavy metals, i.e., lead and cadmium [35,36].

In the applicable legal regulations (European Commission Regulation [38,39]), the permissible cadmium content in the meat of slaughtered animals was set at 0.05 mg/kg, and lead at 0.10 mg/kg. The maximum content of both of these elements in the liver was determined at 0.50 mg/kg.

The analysis of lead content in the meat and a liver of the studied rabbit population in all groups showed properties below the sensitivity of the method. None of the obtained results exceeded the limit value for lead (European Commission Regulation [38]). The obtained results were much lower than the properties obtained by Szkoda et al. [37] for the meat of rabbits from Poland, where the average value of lead in the meat was 0.03 mg/kg (w.w) and 0.17 mg/kg (w.w) in the liver.

Cadmium in the meat and the liver did not exceed the limits in force in the European Union (European Commission Regulation [39]). The concentrations of toxic metals, i.e., Pb and Cd in the liver and meat, were low enough to assume that meat consumption is not a threat to human health.

There are various interactions between metals in a living organism [40,41]. Numerous studies indicate interactions between cadmium and other elements (Cu, Zn, Fe). These occur during their absorption, distribution, and excretion in the body, and on the level of their biological functions in cells [41,42]. There were many studies on the interaction of Cd with Zn, Cu, and Fe, which showed Cd to be poisonous and to affect the homeostasis of these metals, mainly by causing their secondary deficiency [41,43]. Our own research also showed a decrease in microelements, i.e., Zn, Cu, Fe, and Mn, in the liver of the group fed with fodder containing nettle, in which cadmium content was the highest. Moreover, the livers of rabbits fed with nettle had the highest cadmium concentration. This was confirmed by the analysis of interactions between metals, showing a negative correlation between the content of Cd and Zn, Cu, Fe, and Mn in the liver. Similar results were obtained by other authors investigating various animal species [44,45]. An association between Cd and Ni has been faintly described and its exact mechanisms are not well understood. In the liver of rabbits fed with nettle, Ni concentrations increased compared to other groups. In the case of nickel, in contrast to other micronutrients, a positive correlation was observed between its level and the level of Cd in the liver.

The interaction between cadmium and zinc, copper, or iron results from their affinity for metallothionein, and their ability to synthesize this protein in the liver [46,47]. Elements such as zinc and copper protect cells against the toxic effects of cadmium [48]. The protective function is to reduce the accumulation of cadmium and iron inside the cells. The reduction in the content of this metal in cells is due to the antagonism among copper, iron, zinc, and cadmium in cellular transport. Moreover, zinc protects cells induced by cadmium and iron against apoptosis [48,49].

In the literature, there is no information on the correlation among the metals contained in nettle and fenugreek and the content of metals in rabbit tissues. Analyzing the correlations between the concentration of the examined metals in the tissues and their content in herbs obtained in this study, a positive relationship was shown in most cases. Positive correlations were found among Zn, Cu, Ni, and Fe in the nettle and Cd in the liver. In the case of fenugreek, this relationship was observed among Cu, Mn, and Zn in the liver and the meat.

In this study, the content of elements in respect to rabbit gender was also analyzed. Based on the statistical analysis, differences were found in the content of two elements in the liver of female versus male rabbits. Sikora et al. [50] found that the content of copper in female livers was higher than in male livers (5.10 mg/kg and 4.79 mg/kg, respectively). In our research, no statistically significant differences were found in the content of the examined metals in the meat of male and female loin. Sikora et al. [44] found no significant differences between the content of copper and manganese in the muscles of males and females; however, they noted differences in the content of iron (11.93 mg/kg and 17.21 mg/kg, respectively). Similarly, studies by Hermoso de Mendoza García et al. [20] and Bortey-Sam et al. [51] indicated that sex could represent an important source of variation in the bioaccumulation of metals in animals.

The purpose of our work was to investigate whether heavy metals contained in popular and easily available plant additives such as nettle and fenugreek accumulate in rabbit tissues. Our results are the first in the field and will allow us to plan further research on the mechanism of heavy metals in rabbit tissues, including antioxidant enzyme activity and histopathology analysis of tissues. The mechanisms and pathways of toxic effects of heavy metals, mainly cadmium and lead, at the cell and tissue level are not fully understood in biological systems, including rabbits. The results obtained by some authors clearly show that heavy metal toxicity induces a state of oxidative stress reflected in reduced O_2_ consumption at the tissue level in the liver and kidneys. In addition, Cd and Pb induce oxidative stress by reducing the activity of antioxidant enzymes due to changes in gene expression mechanisms and exposure of the living organism to these metals stimulates damage to kidney and liver tissues caused by lipid peroxidation. The disruption of a cellular oxidoreductive balance can lead to serious damage, both on the level of tissues and organs, leading to an impaired function [52,53].

## 5. Conclusions

This experiment helps to explain the interaction between the heavy metal content of nettle and fenugreek and their content in rabbit meat and liver. The meat (m. longissimus lumborum) and liver of rabbits fed with herbal feed contained fewer tested metals than in animals fed with the feed without additives. Concentrations of toxic metals, i.e., Pb and Cd in liver and meat, were so low that meat consumption does not pose a threat to human health. However, more research is needed to determine how the mechanisms and pathways of heavy metal toxicity act on tissue in which these metals are accumulated.

## Figures and Tables

**Table 1 animals-12-00827-t001:** Ingredients and chemical composition of the control and experimental feed (according to feed manufacturer FHP Barbara Ltd.).

Components	C^1^	N^2^	F^3^
Ingredients (% per kg)			
Dicalcium phosphate	0.62	0.62	0.62
Calcium carbonate	0.80	0.80	0.80
Corn	24.50	24.50	24.50
Bran	15.00	15.00	15.00
Wheat	29.58	28.58	28.58
Dried alfalfa	10.00	10.00	10.00
Soybean meal	7.00	7.00	7.00
Sunflower meal	11.00	11.00	11.00
Vitamin-mineral premix	1.50	1.50	1.50
Nettle leaves	0	1	0
Fenugreek seeds	0	0	1
Chemical composition (% per kg)			
Crude protein	16.40	16.55	16.50
Crude fiber	9.22	9.12	9.15
Crude fat	2.70	2.75	2.80
Crude ash	4.84	4.93	4.87
Lysine	0.66	0.66	0.68
Methionine	0.29	0.29	0.30
Calcium	0.77	0.77	0.77
Sodium	0.24	0.24	0.24
Phosphorus	0.63	0.63	0.63
Metabolic energy (MJ/kg)	10.11	10.16	10.19

C^1^—control group, N^2^—diet with nettle leaves, F^3^—diet with fenugreek seeds.

**Table 2 animals-12-00827-t002:** The content of heavy metal in nettle and fenugreek in mg/kg (d.w.).

Heavy Metal	N^1^	F^2^
Zn	25.80 ± 3.10	24.48 ± 2.18
Cu	3.85 ± 0.09	4.45 ± 0.13
Ni	0.57 ± 0.22	0.26 ± 0.56
Fe	380.8 ± 47.67	54.74 ± 1.86
Mn	23.05 ± 0.55	9.14 ± 0.86
Cd	2.29 ± 0.063	0.48 ± 0.057
Pb	u.l.d.	u.l.d.

d.w.—dry weight; u.l.d.—under the limit of detection; N^1^—diet with nettle leaves, F^2^—diet with fenugreek seeds.

**Table 3 animals-12-00827-t003:** The effect of diet on the content of selected heavy metals in rabbit liver in mg/kg (w.w.).

Heavy Metal	C (*n* = 20) ^1^	N (*n* = 20) ^2^	F (*n* = 20) ^3^
Zn	37.34 ^a^ ± 2.54 ^4^	24.19 ^b^ ± 3.18	32.55 ^c^ ± 2.55
Cu	3.65 ^a^ ± 0.26	3.40 ^a^ ± 0.47	4.27 ^b^ ± 0.46
Ni	0.04 ^a^ ± 0.01	0.11 ^b^ ± 0.03	0.06 ^c^ ± 0.01
Fe	82.99 ^a^ ± 10.78	49.70 ^b^ ± 7.88	70.34 ^c^ ± 15.26
Mn	1.93 ^a^ ± 0.21	1.39 ^b^ ± 0.32	1.96 ^a^ ± 0.27
Cd	0.03 ^a^ ± 0.01	0.17 ^b^ ± 0.23	0.02 ^a^ ± 0.01
Pb	u.l.d.	u.l.d.	u.l.d.

w.w.—wet weight; u.l.d.—under the limit of detection; ^1^—control group, ^2^—diet with nettle leaves, ^3^—diet with fenugreek seeds; ^4^—results are presented as mean ± SD. ^a,b,c^—means in rows with different letters are significantly different (*p* ≤ 0.05).

**Table 4 animals-12-00827-t004:** The effect of diet on the content of selected heavy metals in rabbit meat (m. longissimus. lumborum) in mg/kg (w.w.).

Heavy Metal	C (*n* = 20) ^1^	N (*n* = 20) ^2^	F (*n* = 20) ^3^
Zn	5.62 ^a^ ± 0.48 ^4^	5.02 ^b^ ± 0.34	5.51 ^a^ ± 0.36
Cu	0.50 ^a^ ± 0.05	0.35 ^b^ ± 0.04	0.39 ^b^ ± 0.03
Ni	0.22 ^a^ ± 0.04	0.26 ^b^ ± 0.03	0.24 ^b^ ± 0.03
Fe	3.08 ^a^ ± 0.34	2.59 ^b^ ± 0.32	2.56 ^b^ ± 0.33
Mn	0.08 ± 0.01	0.03 ± 0.01	0.02 ± 0.01
Cd	u.l.d.	u.l.d.	u.l.d.
Pb	u.l.d.	u.l.d.	u.l.d.

w.w.—wet weight; u.l.d.—under the limit of detection; ^1^—control group, ^2^—diet with nettle leaves, ^3^—diet with fenugreek seeds; ^4^—results are presented as mean ± SD. ^a,b^—means in rows with different letters are significantly different (*p* ≤ 0.05).

**Table 5 animals-12-00827-t005:** Phenotypic correlations between metals in the liver and meat (m. longissimus lumborum).

Heavy Metal	Zn_liver_	Cu_liver_	Ni_liver_	Fe_liver_	Mn_liver_	Cd_liver_	Zn_l.__l._ ^1^	Cu_l.__l._	Ni_l.__l._	Fe_l.__l._	Mn_l.__l._
Zn_liver_	1	0.36 *	−0.73 *	0.76 *	0.61 *	−0.44 *	0.54 *	0.66 *	−0.44 *	0.36 *	0.57 *
Cu_liver_		1	−0.25	0.17	0.46 *	−0.32 *	0.12	−0.06	−0.14	−0.15	−0.28 *
Ni_liver_			1	−0.65 *	−0.65 *	0.39 *	−0.46 *	−0.52 *	0.40 *	−0.39 *	−0.49 *
Fe_liver_				1	0.46 *	−0.34 *	0.39 *	0.63 *	−0.39 *	0.35 *	0.54 *
Mn_liver_					1	−0.30 *	0.51 *	0.39 *	−0.40 *	0.16	0.26 *
Cd_liver_						1	−0.31 *	−0.17	0.18	−0.24	−0.14
Zn_l.__l._							1	0.38 *	−0.15	0.24	0.25
Cu_l.__l._								1	−0.36 *	0.48 *	0.79 *
Ni_l.__l._									1	−0.24	−0.41 *
Fe_l.__l._										1	0.57 *
Mn_l.__l._											1

*—significant correlation (*p* < 0.05); ^1^
_l.__l._—m. longissimus lumborum.

**Table 6 animals-12-00827-t006:** The phenotypic correlation coefficients among metals in the nettle and liver and rabbit meat.

Heavy Metal	Zn_liver_	Cu_liver_	Ni_liver_	Fe_liver_	Mn_liver_	Cd_liver_	Zn_l.__l._ ^1^	Cu_l.__l._	Ni_l.__l._	Fe_l.__l._	Mn_l.__l._
Zn	0.04	0.08	−0.13	0.28 *	−0.03	−0.11	0.08	0.21	−0.01	0.04	−0.01
Cu	0.31 *	0.20	0.04	0.07	0.09	−0.32 *	0.28 *	0.20	−0.14	−0.01	0.06
Ni	−0.11	−0.11	0.29 *	−0.18	0.05	−0.06	0.02	−0.07	−0.19	0.13	0.11
Fe	−0.11	0.08	−0.08	0.20	−0.06	−0.19	0.03	0.17	−0.01	0.12	0.03
Mn	0.47	0.23	0.09	−0.11	0.20	−0.19	0.28	0.03	−0.17	−0.14	0.05
Cd	0.03	−0.06	0.28	−0.20	0.08	−0.08	0.09	−0.06	−0.19	0.06	0.09

*—significant correlation (*p* < 0.05); ^1^
_l.__l._—m. longissimus lumborum.

**Table 7 animals-12-00827-t007:** The phenotypic correlation coefficients among metals in the fenugreek and liver and rabbit meat.

Heavy Metal	Zn_liver_	Cu_liver_	Ni_liver_	Fe_liver_	Mn_liver_	Cd_liver_	Zn_l.__l._ ^1^	Cu_l.__l._	Ni_l.__l._	Fe_l.__l._	Mn_l.__l._
Zn	−0.06	−0.18	0.02	0.24	0.07	0.68 *	−0.19	0.09	0.19	−0.30 *	−0.12
Cu	0.11	−0.08	0.20	−0.01	0.37 *	0.44 *	0.24	0.37 *	0.06	−0.11	−0.40 *
Ni	0.06	−0.15	0.14	0.10	0.27 *	0.52 *	0.07	0.27 *	0.20	−0.16	−0.28 *
Fe	−0.03	−0.06	−0.10	0.05	0.04	0.63 *	−0.14	0.07	0.01	−0.09	0.38 *
Mn	0.10	−0.10	−0.14	0.38 *	−0.04	−0.04	0.29 *	0.13	0.12	0.13	0.13
Cd	−0.09	0.18	0.07	0.03	−0.40 *	−0.27 *	−0.20	−0.20	−0.24	−0.01	−0.05

*—significant correlation (*p* < 0.05); ^1^
_l.__l._—m. longissimus lumborum.

**Table 8 animals-12-00827-t008:** The content of elements in the liver and meat (m. longissimus lumborum) of females and males in mg/kg (w.w.).

Heavy Metal	Liver		Meat	
	♂(*n* = 30)	♀(*n* = 30)	♂(*n* = 30)	♀(*n* = 30)
Zn	32.05 ± 5.99 ^1^	30.67 ± 3.45	5.36 ± 0.43	5.41 ± 0.52
Cu	3.95 ^a^ ± 0.55	3.60 ^b^ ± 0.49	0.40 ± 0.06	0.43 ± 0.08
Ni	0.07 ± 0.04	0.07 ± 0.03	0.24 ± 0.03	0.24 ± 0.04
Fe	66.21 ± 15.80	69.14 ± 20.12	2.77 ± 0.37	2.72 ± 0.44
Mn	1.88 ^a^ ± 0.37	1.63 ^b^ ± 0.34	0.04 ± 0.03	0.05 ± 0.01
Cd	0.05 ± 0.04	0.09 ± 0.02	u.l.d.	u.l.d.
Pb	u.l.d.	u.l.d.	u.l.d.	u.l.d.

w.w.—wet weight; u.l.d.—under the limit of detection; ^1^—results are presented as mean ± SD. ^a,b^—means in rows with different letters are significantly different (*p ≤* 0.05).

## Data Availability

Data and statistical analysis are available upon request from the corresponding author.

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
