# Peer review of "Effect of a Diet Supplemented with Nettle (Urtica dioica L.) or Fenugreek (Trigonella Foenum-Graecum L.) on the Content of Selected Heavy Metals in Liver and Rabbit Meat"

_animals, 2022, doi:10.3390/ani12070827_

Round 1

Reviewer 1 Report

In my opinion, the manuscript is interesting and contains new information providing accurate data on the content of heavy metals in the liver and muscles of rabbits, and a correlation analysis was performed. I see, however, some minor shortcomings:

- lines 76-78: there is no need here to describe components showed in table 1.

- line 108: why not Tukey test instead of Tukey-kramer test? Don’t you have equal n?

- lines 119-121: this sentence should be put in Introduction section.

- table 2: this table should belong to M&M section (look at the title of the manuscript).

- line 152: As expected, …

- lines 156-157: check this sentence for grammar.

- line 176: Analysis showed…

- lines 201-208: Try to combine this paragraph with your results so that it is part of the discussion

- line 216: In our research, …

- line 232-233: why are you adding the phrase ,,(i.e. industrial livestock)” here?

- lines 234-235: ,,in both the liver of rabbits”? You also mean meat?

- lines 235-241: This assumption can be made without looking at your results so this sentence fits into the Introduction section (not in Conclusions). Perhaps it would be appropriate to provide a bit more detail of your results in this section?

Author Response

Krakow, 23.02.2022

Dear Reviewer,

We would like to thank you for all your comments and suggestions regarding our manuscript. We have made all the suggested changes to the text. We have corrected language errors. We separated the results from the discussion. We have changed our conclusions. Table 2 has been moved to the M&M section.

Reviewer 2 Report

GENERAL COMMENT:

I consider this work is within the scope of “Animals”. It contains information useful in a field in which available information is of interest to improve knowledge on rabbit feeding and nutrition and meat quality. Overall, it is well organised. However, I indicate several flaws found in the manuscript. I indicate these flaws below and in a commented PDF file I have uploaded.

Along the entire manuscript: Do not type in italics “L.” from Linneo in latin names of the plants.

ABSTRACT:

Line 4: indicate: “10 males and 10 females”.

Line 4: type “ad libitum” in italics.

Line 8: replace “affect” with “affected”.

INTRODUCTION:

Insert spaces where indicated.

MATERIALS AND METHODS:

Line 68: insert “(Poland)”.

Line 76: type “ad libitum” in italics.

Line 79: if the complete mixture used for N and F treatments, do not write “a” mixture, but “the same mixture” (or similar wording).

Line 90: according to the journal style for citations, write: "by Blasco et al [22]".

Line 97: write correctly chemical formulas.

RESULTS AND DISCUSSION:

Line 114: replace “gender” with “sex”.

Line 154: replace commas where indicated.

Line 220: Rewrite as: "Hermoso de Mendoza et al. [20] and Bortey-Sam et al. [45]".

REFERENCES SECTION:

In general terms, this section is well organised and adjusted to the style of the journal for references. However, some improvement is possible. For example:

Type in italics the Latin names of the plants.

TABLES:

Tables need to be interpreted independently of the manuscript text. Therefore, some improvement is needed:

Table 1: As the mineral content of the premix can influence meat quality, add its composition.

Table 1: Indicate whether this composition is ad-feed of dry matter basis.

Table 2: Add a footnote in the table to explain what does "d.w." mean.

Tables 3, 4 and 6: Add a footnote in the table to explain what does "w.w." means.

Tables 3, 4 and 6: Indicate whether data are mean±ET or mean±SD.

Author Response

Krakow, 23.02.2022

Dear Reviewer,

We would like to thank you for all your comments and suggestions regarding our manuscript. We have made all the suggested changes to the text. We have corrected language errors. We have corrected the citation style. The Latin text is in italics. We have added explanations to Tables 1, 2, 3 and 4.

Reviewer 3 Report

The authors investigated the effect of diet supplemented with nettle (Urtica dioica L.) or fenugreek (Trigonella foenum-graecum L.) on the content of selected heavy metals in the liver and rabbit meat. They designed three treatments to test the effect of these supplements on heavy metal levels in liver and muscle. This manuscript (MS) was clearly written and easy to understand. Although they covered heavy metal contents, still some gaps are here in terms of provided data. However, some major issues significantly compromised the quality of this MS.

Major comments:

  • First, the manuscript needs to be edited by a native English speaker to improve the language of the MS and fix errors. I mentioned some of them, but still, more effort is required to be made.
  • They did not provide adequate data and evidence for supporting their hypothesis. The most important part of this research is the antioxidant and/or liver enzymes. For publishing in a high-quality journal like this, more data should be provided. Further, the metal contents of diets should be reported and then can be correlated with liver and muscle data.

However, I have touched on some more points that can contribute to the improvement of this MS.

Minor comments

Abstract

  • Line 4, please revise this; it is not clear. I think you meant to duplicate?.
  • Line 6, for how long?
  • Line 11, revise this part.
  • Line 12, which tissues.
  • It is better to start the abstract with a sentence about why did you analyze this work or something about the importance of herbal medicines.
  • Here and elsewhere, report P uppercase and italic (P<0.05).
  • Throughout the MS, if there is no significant difference, no need to report P-value.

Introduction:

  • Well-developed introduction and included a clear fellow and relevant points.
  • Line 21, in culinary or cultural contexts, veal is often considered white meat, but it actually is red meat. It is better to replace it with fish.
  • Line 25, revise.
  • Line 40-41, revise.
  • Line 43-46, please recheck the numbers, too many protein contents.
  • Line 56, also include some antinutritional factors that cause problems in the digestive system and intestine. Please mention to them as well.
  • Please review the studies that used herbs in rabbits in a separate paragraph.
  • Please mention the novelty of your work in the last paragraph of the introduction.
  • Here and throughout the MS, please first mention the common name plus scientific name, and for the rest of the MS, just report the common name. Many errors from this point can be seen.

Material and methods

  • Well-organized section. Clear fellow and all required details were provided.
  • Line 73-74, please revise it.
  • Line 65, I can not see how long the period of the experiment was. I suggest adding growth data as well to this MS.
  • The content of heavy metal in the experiment should be reported. Otherwise, we can not see how bioaccumulation in tissues occurred.
  • For each analysis, please clarify how many animals were taken for each analysis.

Results and discussion

  • Well-written section, all necessary things have been covered, but some essential data should be added.
  • Please report the growth data, Further the antioxidant parameters in at least the liver. It is required, and there is another piece of the puzzle to see how heavy metals work.
  • 119, if you mean human food is fine, but if you meant rabbit, feed is a better word.
  • Please combine tables 2 and 3 into one table.
  • Line 140, readers have no idea whether these levels in muscle is harmful to human or not. Please provide the standards and report which heavy metal in muscle is upper than the safe level for a human.
  • Line 161, please notice that some of the sentences you brought here are not relative. In your study, you did not see any high level of some of them. As I said in the above comment, when you compare the levels in muscle with international standards, you will understand which one can be harmful. Then, you can discuss it.
  • Line 166, ok you mentioned here. Please find the standard levels of all mentioned metals based on European Commission Regulation or other resources.
  • Line 201, it is a good idea to have a correlation between the metal contents in diets with liver and muscle.
  • Line 217, revise.
  • Please check the tables; you reported two N, which I think one of them should be F.
  • In table 5 and other tables, I can see many errors. Please double check all of them.
  • Table 5, please report the data that correlation was significantly based on 0.05 and 0.01. Please see the correlation tables in other good papers to get an idea.
  • Check table 6; I can see the error in the subset. “a” should be “b”?
  • Line 231, please revise; it is not clear and checks the parenthesis.

Best regards

Author Response

Krakow, 23.02.2022

Dear Reviewer,

We would like to thank you for all your comments and suggestions regarding our manuscript. We have made the suggested changes to the text. We have corrected language errors. We separated the results from the discussion.

Abstract:

We have added the missing information in this part of the text.

Introduction:

Line 25, revise.

Answer: Answer: We changed the text.

Line 40-41, revise.

Answer: We changed the text.

Line 43-46, please recheck the numbers, too many protein contents.

Answer: The  literature show that the protein content in fenugreek seeds is 20-30%.

Line 56, also include some antinutritional factors that cause problems in the digestive system and intestine. Please mention to them as well.

Answer: We have added the missing information in this part of the text.

Material & methods:

We have added the missing information in this part of the text.

Line 73-74, please revise it.

Answer: In the experiment, we used 10 does. We have corrected our mistake.

The content of heavy metal in the experiment should be reported. Otherwise, we cannot see how bioaccumulation in tissues occurred.

Answer: The source of heavy metals in this experiment were herbs administered in food, i.e. fenugreek and nettle. The content of heavy metals in these herbs is given in Table 2.

For each analysis, please clarify how many Animals were taken for each analysis.

Answer: The all rabbits were slaughtered on.... This information has been added in Materials and methods. Samples of meat and liver were collected from each rabbit for heavy metal analysis by atomic absorption spectroscopy. This information has been added in Materials and methods.

Results and discussion:

We have added the missing information in this part of the text.

Please report the growth data, Further the antioxidant parameters in at least the liver. It is required, and there is another piece of the puzzle to see how heavy metals work.

Answer: The authors agree with the Reviewer that antioxidant parameters studies provide information on the mechanism of heavy metal action. These studies take a long time, therefore we were not able to repeat them in such a short time (10 day - response to reviews). Our results are the first in the field and will allow us to plan further research on mechanism of heavy metals in rabbit tissues, including antioxidant enzyme activity and histopathology analysis of tissues.

Line 140, readers have no idea whether these levels in muscle is harmful to human or not. Please provide the standards and report which heavy metal in muscle is upper than the safe level for a human.

Line 161, please notice that some of the sentences you brought here are not relative. In your study, you did not see any high level of some of them. As I said in the above comment, when you compare the levels in muscle with international standards, you will understand which one can be harmful. Then, you can discuss it.

Line 166, ok you mentioned here. Please find the standard levels of all mentioned metals based on European Commission Regulation or other resources.

Answer: The recommended daily consumption of metals such as: zinc, copper, iron, manganese, nickel in an adult human is 12.7 mg/day, 1.6 mg/day, 6 mg/day, 3 mg/day, 150 μg/day, respectively (EFSA 2013, 2014, 2015a,b,c). The consumption of rabbits in the world depends on in culinary traditions of countries. Overall, the consumption of rabbit meat in the EU is 0.5 kg per person a year. Taking into account the average content of these micronutrients in the meat of the tested rabbits, the average annual consumption in the EU and the absorption from food on the level of about a few percent may suggest that their content does not pose a risk to the health of the consumer.

- This information has been added in Discussion.

EFSA 2013. Scientific Opinion on Dietary Reference Values for manganese. EFSA Journal 11(11):3419

EFSA 2014. Scientific Opinion on Dietary Reference Values for zinc. EFSA Journal 12(10):3844

EFSA 2015a. Scientific Opinion on Dietary Reference Values for copper. EFSA Journal 13(10):4253

EFSA 2015b. Scientific Opinion on Dietary Reference Values for iron. EFSA Journal 13(10):4254

EFSA 2015c. Scientific Opinion on Dietary Reference Values for nickel. EFSA Journal 13(2):4002

Please combine tables 2 and 3 into one table.

Answer: Table 2 has been moved to the M&M part as suggested by one of the Reviewers.

Line 201, it is a good idea to have a correlation between the metal contents in diets with liver and muscle.

Answer: We have added Table 6 and 7 with correlations among the metal contents in diet with liver and muscle.

Please check the tables; you reported two N, which I think one of them should be F.

In table 5 and other tables, I can see many errors. Please double check all of them.

Table 5, please report the data that correlation was significantly based on 0.05 and 0.01. Please see the correlation tables in other good papers to get an idea.

Check table 6; I can see the error in the subset. “a” should be “b”?

Line 231, please revise; it is not clear and checks the parenthesis.

Answer: We have added explanations to Tables 2, 3, 4, 5, 6, 7 and 8. We have corrected errors in the tables.

Round 2

Reviewer 3 Report

The authors have not improved the quality of the MS. The fundamental problem related to data has not been solved, and from my idea, this MS can not go to further steps.

Author Response

We wish to inform you that we are not able to meet the most important requirement of the Reviewer, it is the determination of antioxidant enzymes in the liver and rabbit meat. The authors agree with the Reviewer that antioxidant parameters studies provide information connected with the mechanism of heavy metal action. Planning this study, we did not know any results of other studies that show the accumulation of heavy metals in the meat and liver of rabbits fed with nettle and fenugreek. Therefore, we undertook such studies. Our results are the first in the field and will allow us to plan further research on mechanism of heavy metals in rabbit tissues, including antioxidant enzyme activity and histopathology analysis of tissues. To determine these enzymes, we would have to carry out our experiment again. These studies were done a long time ago (50 days) therefore, we are unable to retake them in such a short time (10 day - response to reviews).

In the meantime, we would like to inform you that we made the changes suggested in the text.

In the abstract tere is information why this research was done, we suplemented the methodology in more detail and added the significance level P<0.05.

In the Material and Methods section, in accordance with the suggestions of Reviewer, we corrected information on the number of rabbits and also added information including the duration of the experiment, the number of animals from which samples were taken.

In the part of the results, we counted the the correlations, included in tables 6 and 7. We also corrected the description of the other tables and checked the data contained in them. We separated the Results from the Discussions. The suggested information was addend to the discussion part.

Yours sincerely,

Authors'
